## Reply

analytical chemistry/spectroscopy/computer modelling and simulation

factorial, fluconazole, itraconazole, terbinafine, isocratic, HPLC

**Author for correspondence:**
Aya Roshdy
e-mail: yoyafouda94@gmail.com

This article has been edited by the Royal Society of Chemistry, including the commissioning, peer review process and editorial aspects up to the point of acceptance.

# Factorial design-assisted reversed phase-high performance liquid chromatography method for simultaneous determination of fluconazole, itraconazole and terbinafine

Aya Roshdy[1,2], Heba Elmansi[1], Shereen Shalan[1] and Amina El-Brashy[1]

[1]Department of Pharmaceutical Analytical Chemistry, Faculty of Pharmacy, Mansoura University, Mansoura 35516, Egypt
[2]Department of Pharmaceutical Chemistry, Faculty of Pharmacy, Horus University-Egypt, New Damietta, Egypt

AR, 0000-0002-7195-9361; HE, 0000-0002-3953-7169; SS, 0000-0002-1468-1367

A $2^3$ full factorial design model was used for the development of a new high performance liquid chromatography method with UV detection to estimate three antifungal drugs simultaneously. Fluconazole (FLU), itraconazole (ITR) and terbinafine (TRH) are co-administered for severe fungal infections. They have been determined using MOS-1 Hypersil $C_{18}$ column and an isocratic eluent; methanol 95% and phosphate buffer 5% with 0.001% triethylamine. The pH was adjusted to 7, and the flow rate was 0.7 ml min$^{-1}$. The three drugs were separated within less than 7 min at 210 nm. The developed method gave a linear response over 5–80 µg ml$^{-1}$, 5–50 µg ml$^{-1}$ and 1–50 µg ml$^{-1}$ for FLU, ITR and TRH, respectively. It showed detection limits of 0.88, 0.29 and 0.20 µg ml$^{-1}$ and quantification limits of 2.66, 0.88 and 0.60 µg ml$^{-1}$ for the three drugs, respectively. The design of the experiment facilitated the optimization of different variables affecting the separation of the three drugs. The sensitivity of the designed method permitted the simultaneous estimation of ITR and TRH in spiked human plasma successfully.

**Figure 1.** The structural formulae of the studied drugs: (*a*) fluconazole, (*b*) itraconazole and (*c*) terbinafine HCl.

# 1. Introduction

Increasing resistance to conventional antifungal treatment has required that novel strategies of therapy should be introduced. Recent studies have revealed that some combinations may be effective in some resistant species of fungi [1]. This work deals with three commonly described antifungal drugs which are:

— fluconazole (FLU) is 2-(2,4-difiuorophenyl)-1,3-bis (IH-1,2,4-triazol-1-yl) propan-2-ol. It is slightly soluble in water and freely soluble in methanol [2];
— itraconazole (ITR) is 4-[ 4-[ 4-[ 4-[[ *cis*-2-(2,4-dichlorophenyl)-2-(1H-1, 2A-triazol-1-ylmethyl)-1, 3-dioxolan-4-yl] methoxy] phenyl] piperazin-1-yl] phenyl] -2-[ (1RS)-1-methylpropyl]-2,4-dihydro-3H-1,2,4-triazol-3-one [2]; and
— terbinafine (TRH) is [(2E)-6,6-dimethylhept-2-en-4-yn-1-yl] (methyl)(naphthalen-1-ylmethyl) amine hydrochloride [2].

Various methods have been investigated for the three drugs determination individually including spectrophotometry for FLU [3–6], for ITR [7–9] and for TRH [10–13], spectrofluorometry for FLU [14], for ITR [15] and for TRH [16] and chromatography methods for FLU [17–21], for ITR [22–27] and for TRH [28–31].

Different analytical methods were reported for the determination of combination of itraconazole and terbinafine as they are used in the treatment of severe fungal infection [32–34]. Also, a combination of fluconazole and itraconazole was determined spectrophotometrically as they are used to treat *Candida* isolates [35]. The combination of these three antifungal drugs (chemical structures are illustrated in figure 1) is recommended for the treatment of *Aspergillus*, *Candida*, Mucorales species and against fluconazole-resistant *Candida* isolates and itraconazole-resistant *Aspergillus* strains with minimal side effects and high efficacy [36].

In this study, a novel high performance liquid chromatography method is designed for the simultaneous analysis and quantitation of fluconazole, itraconazole and terbinafine, to our knowledge for the first time. The design of experiment (DOE) has been involved as an efficient methodology to test the influence of multiple factors which affect certain responses and the interaction between them with a low number of trials. Hence, it also participated in a decreasing amount of organic solvents and chemicals consumed during the study.

# 2. Experimental measures

## 2.1. Apparatus, materials, solvents and reagents

— Shimadzu Prominence HPLC system (Shimadzu Corp., Kyoto, Japan) with an LC-20 AD pump, DGU-20 A5 degasser, CBM-20A interface and SPD-20A UV/VIS detector. Rheodyne injector valve and 0.45 µm membrane filters (Millipore, Cork, Ireland).

— All the chromatographic data obtained were manipulated using Perkin Elmer TM Series Software.

— Full factorial design and statistical analysis were carried out using MINITAB® statistical software (release 16 for windows, State College, Pennsylvania, PA, USA).

— Consort NV P-901 pH Meter (Belgium) was used to adjust pH.

— FLU was kindly supplied by Amoun Pharmaceutical Co. (El- Obour City, Cairo, Egypt). The purity percentage of FLU was 99.3%.

— TRH was kindly donated by Novartis Pharma AG, Basle, Switzerland). The purity percentage of TRH was 99.2%.

— ITR pure sample was purchased from Multi Apex Pharmaceutical Industries S.A.E, Badr City, Egypt. The purity percentage of ITR was 99%.

— Organic solvents (HPLC grade) were obtained from Sigma-Aldrich (Germany).

— Orthophosphoric acid (85%, w/v) was purchased from RiedeldeHäen (Seelze, Germany).

— Sodium dihydrogen phosphate and sodium hydroxide were obtained from ADWIC Co. (Cairo, Egypt).

— Trietheylamine (greater than or equal to 99.5%) was obtained from Sigma-Aldrich (Germany).

Commercial dosage forms were obtained from pharmacies in Egyptian market including:

— Flucoral® capsules containing 150 mg fluconazole, product of Alfa Cure Pharmaceuticals Company, Cairo, Egypt;

— Itrapex® capsules containing 100 mg itraconazole, product of Global Napi Pharmaceuticals, Cairo, Egypt;

— Lamisil® tablets containing 250 mg terbinafine HCl, product of Novartis, Cairo, Egypt produced by Al-Andalus, Cairo, Egypt; and

— human plasma samples were provided by Mansoura University Hospitals and kept frozen at −20°C until used.

## 2.2. Standard solutions and mobile phase

Samples of each of FLU, ITR and TRH were accurately weighed and dissolved in methanol in 100 ml volumetric flasks to yield 100 µg ml$^{-1}$ standard solutions.

The mobile phase consists of methanol 95% and phosphate buffer 5% with 0.001%, v/v triethylamine adjusted at pH 7. The solution was subjected to ultrasonication and filtration through 0.45 µm membrane filters.

## 2.3. Procedures

### 2.3.1. Constructing the calibration graphs

Aliquots of each of FLU, ITR and TRH standard solutions equivalent to final concentrations of 5.0–80.0, 5.0–50.0 and 1.0–50.0 µg ml$^{-1}$ for the three drugs, respectively, were measured and transferred carefully into series of 10 ml volumetric flasks, completed with the mobile phase and mixed. Under optimum chromatographic sittings, 20 µl aliquots were injected in triplicate and the mobile phase flowed at a rate of 0.7 ml min$^{-1}$. The average peak area ($y$) was plotted against the final concentration of the drug ($c$) in µg ml$^{-1}$ to construct the calibration curve for the analytes.

### 2.3.2. Analysis of pharmaceutical preparations

Separately, 10 capsules of each of Flucoral® and Itrapex® were evacuated and 10 tablets of Lamisil® were ground, mixed and weighed precisely. A certain weight of each powdered drug was accurately transferred into a 100 ml volumetric flask to give concentrations equivalent to one capsule or tablet (250, 100 and 250 mg FLU, ITR and TRH, respectively), then about 40 ml of methanol was added. Each flask was subjected to sonication for 30 min, then the volumes were completed to 100 ml using methanol. The solutions were then filtered to get clear solutions. Working standard solutions were analysed as discussed in 'Constructing the calibration graphs'.

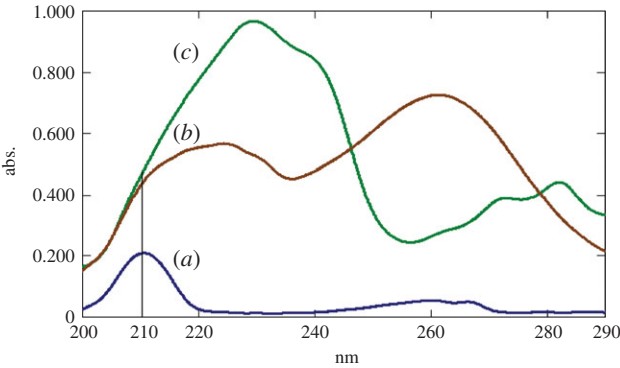

**Figure 2.** Absorption spectra of the three drugs: (*a*) FLU, (*b*) ITR and (*c*) TRH (10 µg ml$^{-1}$ each).

**Table 1.** The $2^3$ experimental factorial designs and their dependent responses for reversed phase (RP)-HPLC-UV separation of FLU, ITR and TRH mixture. (pH, aqueous mobile phase pH (low level 5 and high level 7). MeOH%, MeOH% (v/v) in organic mobile phase (low level 85% and high level 95%). Flow rate (low level 0.7 and high level 1). Rs1, resolution between FLU and ITR. Rs2, resolution between ITR and TRH.)

| dependent responses | | experimental factorial design | | | dependent responses | | | |
|---|---|---|---|---|---|---|---|---|
| std order | run order | pH (*A*) | %MeOH (*B*) | flow rate (*C*) | NTP of TRH | tailing of TRH | Rs1 | Rs2 |
| 8 | 1 | 7 | 95 | 1.0 | 3416 | 1.55 | 2.26 | 4.39 |
| 3 | 2 | 5 | 95 | 0.7 | 2290 | 2.67 | 2.59 | 5.52 |
| 6 | 3 | 7 | 85 | 1.0 | 2907 | 2.30 | 8.76 | 11.90 |
| 7 | 4 | 5 | 95 | 1.0 | 2095 | 2.53 | 2.72 | 6.32 |
| 2 | 5 | 7 | 85 | 0.7 | 2672 | 2.49 | 8.95 | 12.35 |
| 4 | 6 | 7 | 95 | 0.7 | 5198 | 1.60 | 3.00 | 6.45 |
| 5 | 7 | 5 | 85 | 1.0 | 2088 | 2.22 | 7.94 | 9.61 |
| 1 | 8 | 5 | 85 | 0.7 | 2262 | 2.46 | 7.95 | 9.84 |

**Table 2.** Response optimization of $2^3$ factorial design for RP-HPLC-UV separation of FLU, ITR and TRH mixture.

| | | | | | | | optimum solution: pH = 7, MeOH % = 95%, flow rate = 0.7 | |
|---|---|---|---|---|---|---|---|---|
| | parameters | | | | | | composite desirability (*D*) = 1 | |
| response | goal | lower | target | upper | weight | importance | predicted responses | desirability (days) |
| NTP | maximum | 2088 | 5198 | 5198 | 1 | 1 | 5198 | 1 |
| tailing | minimum | 1.55 | 1.6 | 2.67 | 1 | 1 | 1.6 | 1 |
| Rs1 | target | 2.26 | 3 | 8.95 | 1 | 1 | 3.0 | 1 |
| Rs2 | minimum | 4.39 | 6.5 | 12.35 | 1 | 1 | 6.45 | 1 |

### 2.3.3. Analysis in spiking plasma samples

Using the proposed method, both ITR and TRH can be determined in human plasma. Aliquots of 1 ml of human plasma were transferred into a set of fixed capped tubes then increasing volumes of ITR and TRH standard solutions (final concentration reached: 5–13 µg ml$^{-1}$ for ITR and 0.5–2 µg ml$^{-1}$ for TRH) were added and mixed well. Methanol was used to complete the volume of each tube to 5 ml. Samples were subjected to vortex mixing for 15 s, then centrifugation at 3500 rpm for 30 min confirming the total separation of the drugs from plasma contents. The clear layer was filtered using syringe filters. Aliquots from the filtrate were accurately transferred into a 10 ml set of volumetric flasks and analysed by performing a blank experiment. The plasma content from the studied drugs is determined using regression equations.

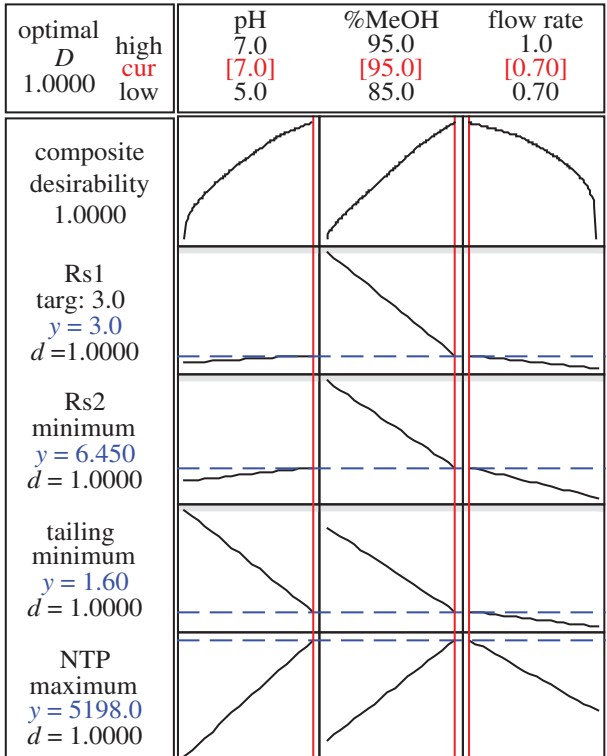

**Figure 3.** A $2^3$ full factorial design optimization plot.

# 3. Results and discussion

## 3.1. Optimizing the method

### 3.1.1. Choice of column

Three columns were investigated, which are: the Nucleosil 100-5 Phenyl column (250 mm × 4.6 mm i.d., 5 μm particle size), the Shimadzu VP-ODS $C_{18}$ column (250 mm × 4.6 mm i.d., 5 μm particle size) and the Thermo scientific ODS Hypersil $C_{18}$ column (250 mm × 4.6 mm i.d., 5 μm particle size)

The latter was preferred as it gave distinct peaks with good resolution and short analysis time while the other columns gave longer analysis time.

### 3.1.2. Wavelength detection

Based on the UV spectra of the three drugs, 210 nm was selected as an optimum UV detection wavelength because they all show high absorbance at this wavelength, as shown in figure 2.

### 3.1.3. Mobile phase composition, pH and flow rate

Organic solvents as methanol, ethanol and acetonitrile were examined; it was found that the optimum one was methanol, as acetonitrile and ethanol induced overlapping of the two peaks of fluconazole and itraconazole. According to the prior trials, decreasing the ratio of methanol resulted in a major increase in the retention time of TRH, so 85–95% of methanol was investigated for the study.

The optimization of chromatographic conditions 'univariate optimization' is a complex process as it requires many experiments to achieve the optimum conditions. The study includes one variable at a time, while others remain constant which is time-consuming.

A full factorial design is a type of DOE 'multivariate optimization' which allows us to investigate the effect of all the factors with a simultaneous variation of them and to watch the responses encountered from independent factors and the interactions between those factors [37]. In this study, for the optimization of the chromatographic condition, $2^3$ full factorial design was applied as it included two levels and three independent factors. The most important factors affecting the HPLC method were per cent of methanol,

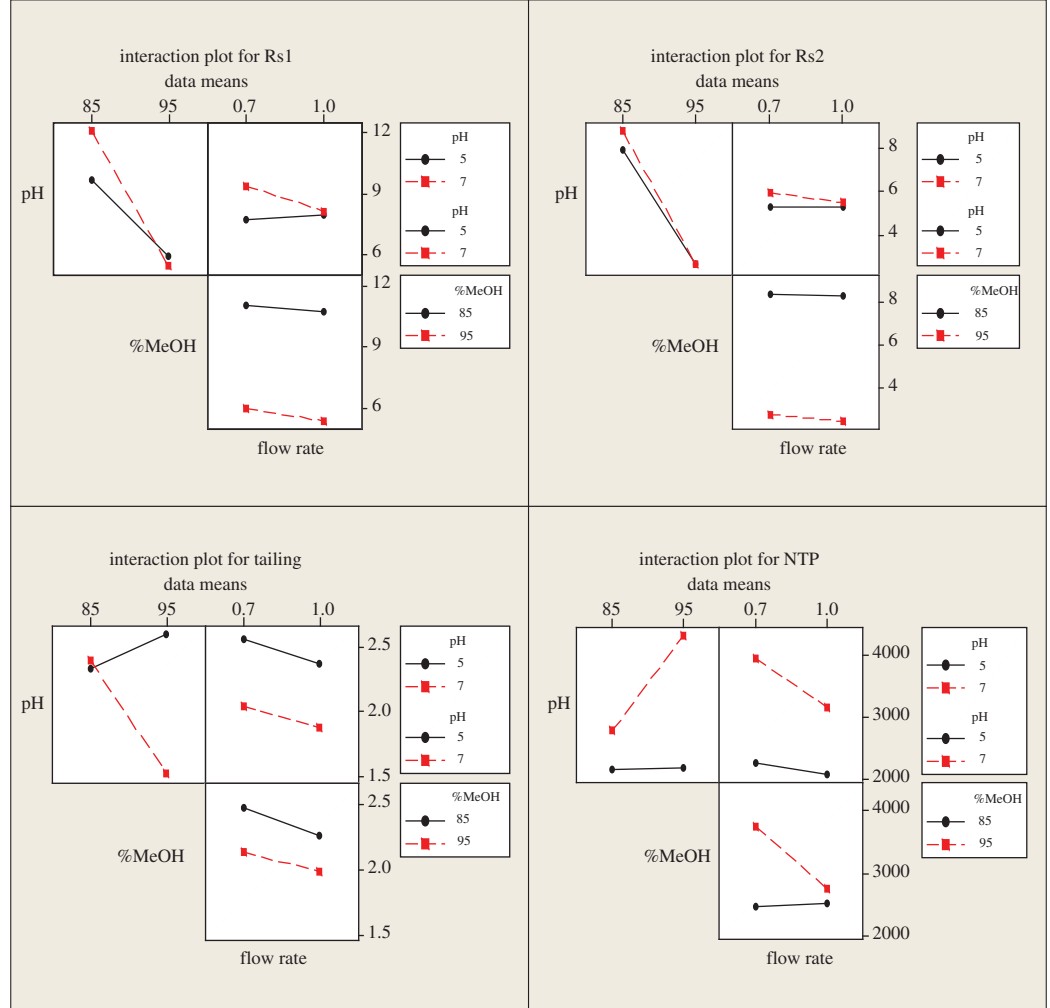

**Figure 4.** The $2^3$ full factorial design full interaction plots for chromatographic responses by data means type.

flow rate and buffer pH and their optimization was carried out by DOE. Two different ratios of methanol such as 85 and 95 were studied; the pH of sodium hydrogen phosphate selected to be studied were 5 and 7. These selected pHs were reported in the analysis of acidic and basic drugs in a reversed phase (RP) system and the flow rates of 0.7 and 1 ml min⁻¹ were chosen. Consequently, the two levels were (−1) for the lower level and (+1) for the higher level and the three independent factors were per cent of methanol (A), pH of buffer (B) and flow rate (C) [38].

The $2^3$ full factorial design suggested eight experiments to analyse the interaction of each level on the responses, which were the resolution between fluconazole and itraconazole (R1), resolution between itraconazole and terbinafine (R2), tailing of terbinafine peak (R3) and number of the theoretical plate of TRH USP (NTP) (R4). The two levels, independent variables and dependent variables, are illustrated in tables 1 and 2.

The significance of independent factors was evaluated by means of the estimated Fisher statistical test for variance analysis (ANOVA) model [39], which is applied on the responses to study the effect of these independent factors on the responses and the interactions between them. The polynomial equation for the experimental design with three factors is given below:

$$R = \beta_0 + \beta_1 A + \beta_2 B + \beta_3 C + \beta_2 AB + \beta_2 AC + \beta_2 BC + \beta_2 A^2 + \beta_2 B^2 + \beta_2 C^2,$$

where R is the response, β is the regression coefficients and A, B and C represent per cent of methanol, pH of buffer and flow rate, respectively.

To ensure that the optimum conditions are obtained, MINITAB response optimizer calculates the composite desirability (D) which evaluates if the responses are in their acceptable limits and it ranges from zero to one. Zero is not accepted as it means that many of the responses are out of their accepted limits, while one means that the condition reached is optimum, so its value is better to be one or near one (table 2).

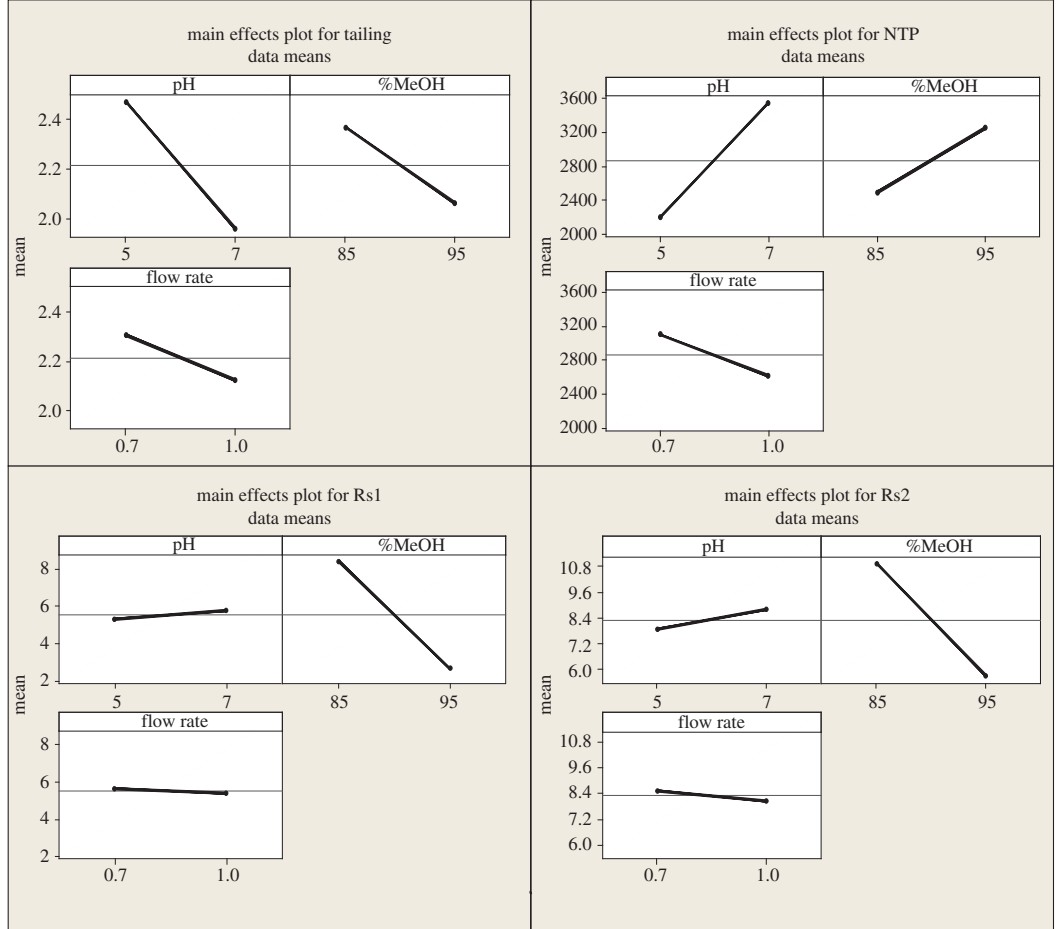

**Figure 5.** The $2^3$ full factorial design main effect plots for chromatographic responses by data means type.

A full factorial design resulted in the optimum solution; hence, the optimization plot (figure 3) shows how the composite desirability and responses are affected by the three factors and the interaction between them to reach the optimum condition.

The factorial design can also provide interaction plots (figure 4), which show that increasing methanol and pH and decreasing flow rate lead to increasing NTP and Rs1 and minimizing tailing and Rs2 which is confirmed by the main effect plots (figure 5). Pareto charts (figure 6) show that methanol% in the mobile phase (B) has a major effect on the chromatographic performance (Rs1 and Rs2) and has a statistically significant effect for a 95% confidence level, while pH (A) has the major effect on NTP. However, it is not potentially effective for a 95% confidence level. pH and methanol% (AB) have the strongest effect on tailing response.

Finally, the mobile phase was selected in the 95 : 5 ratio of methanol : buffer with 1 ml of 0.5%TEA. The final flow rate of 0.7 ml min$^{-1}$ is based on DOE. The UV detector was set at 210 nm to allow the detection of drugs in the samples.

The results of the optimization are compared with the pharmacopeial values [40], as shown in table 3 and figure 7, where:

$$\text{number of theoretical plates } (N) = 5.54\left(\frac{t_R}{W_{h/2}}\right)^2,$$

$$\text{resolution } (R_s) = \frac{2\Delta t_R}{W_1 + W_2},$$

$$\text{capacity factor } (k') = \frac{t_R - t_m}{t_m},$$

$$\text{selectivity } (\alpha) = \frac{k'_2}{k'_1},$$

and
$$\text{tailing factor } (T) = \frac{W_{0.05}}{2d}.$$

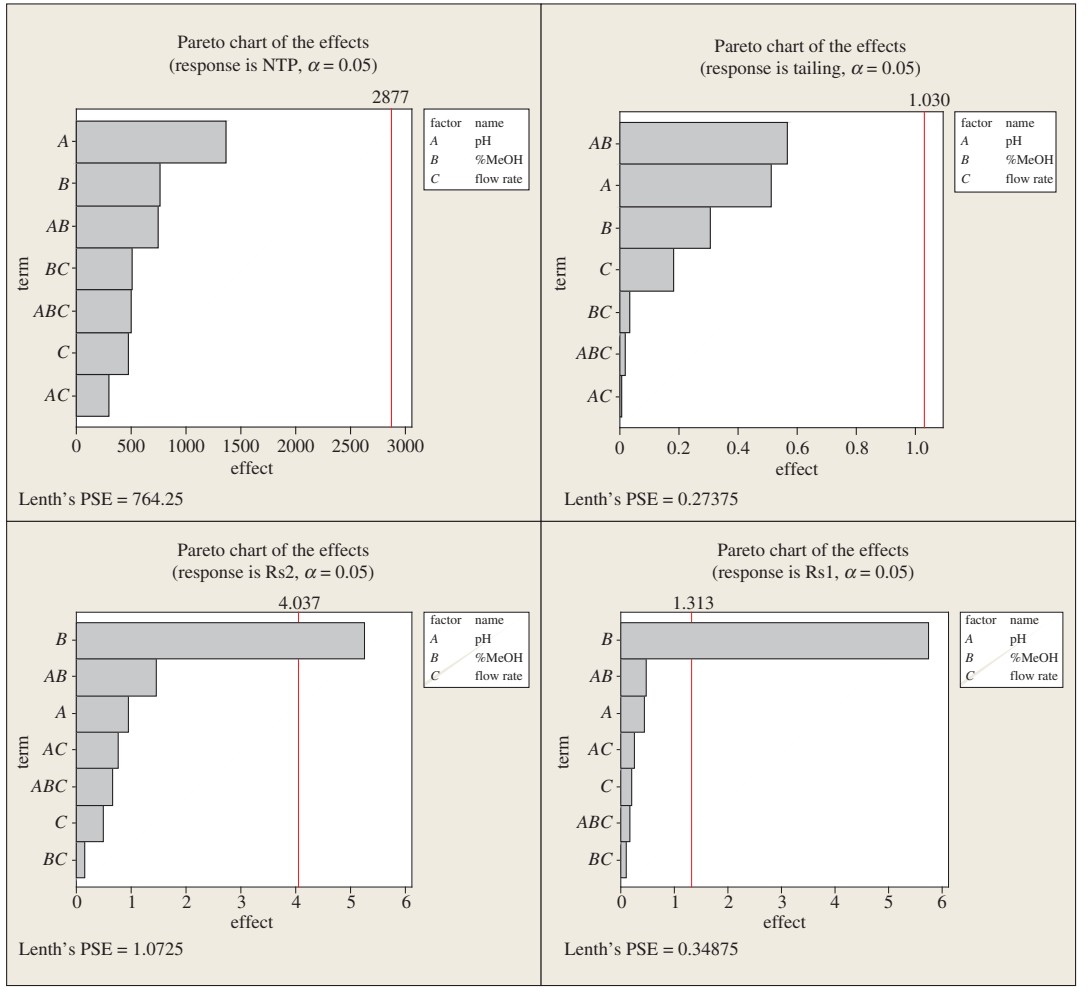

**Figure 6.** The $2^3$ full factorial design Pareto charts of the effects on the chromatographic responses at $\alpha = 0.05$.

**Table 3.** Analytical performance data for the determination of FLU, ITR and TRH by the proposed method.

| parameter[a] | FLU | ITR | TRH |
|---|---|---|---|
| no. of theoretical plates, $N$ | 743 | 1159 | 2568 |
| capacity factor, $k'$ | 0.19 | 0.61 | 2.11 |
| selectivity factor, $\alpha$ | 3.12 | | 3.48 |
| resolution, $R_s$ | 2.23 | | 6.45 |
| retention time ($t_R$), min | 2.32 | 3.13 | 6.07 |
| tailing factor ($T$) | 1.80 | 1.80 | 1.50 |

[a]Reference values are: $k'$: 0–10, $\alpha > 1.0$, $R_s > 1.5$ and $T$: 0.9–1.2 [40].

## 3.2. Validation parameters

Validation was carried out as guided by the International Conference on Harmonization (ICH) Q2R1 [41]. The linearity and ranges for the three drugs were investigated by the regression equations after analysis of six concentrations for each of the three drugs. The suggested HPLC method was applied over the ranges of (5–80, 1–50 and 1–50 µg ml$^{-1}$), respectively, to pure samples of FLU, ITR and TRH, as shown in table 4. The peak area ($y$) was plotted against the concentration ($c$) and correlation

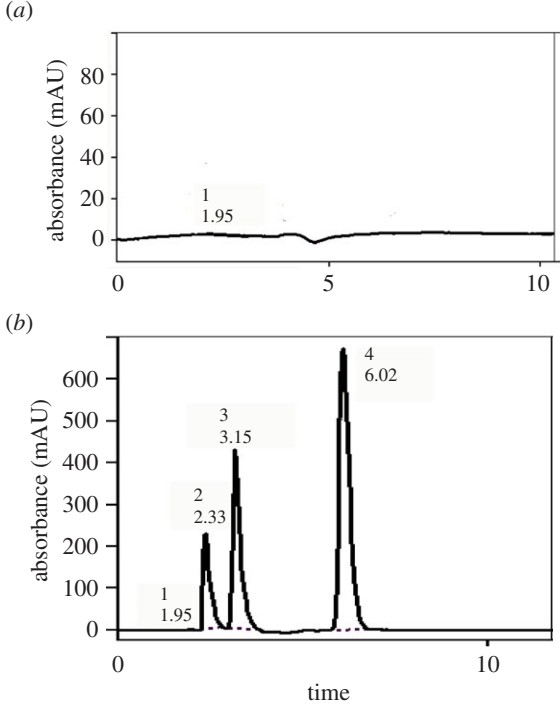

**Figure 7.** Typical chromatogram of the studied drugs under the described chromatographic conditions. (*a*) Sample blank, (*b*) 1, solvent front; 2, FLU (80 µg ml$^{-1}$); 3, ITR (50 µg ml$^{-1}$); and 4, TRH (50 µg ml$^{-1}$).

**Table 4.** Analytical performance data for the determination of the FLU, ITR and TRH by the proposed HPLC method. ($S_{y/x}$, standard deviation of the residuals; $S_a$, standard deviation of the intercept of regression line; $S_b$, standard deviation of the slope of regression line. % error = RSD% / $\sqrt{n}$.)

| parameter | FLU | ITR | TRH |
|---|---|---|---|
| linearity range (µg ml$^{-1}$) | 5.0–80.0 | 5.0–50.0 | 1.0–50.0 |
| intercept (*a*) | −75.07 | 135.96 | 62.51 |
| slope (*b*) | 38.35 | 82.28 | 229.14 |
| correlation coefficient (*r*) | 0.9999 | 0.9999 | 0.9999 |
| s.d. of residuals ($S_{y/x}$) | 14.07 | 9.39 | 19.64 |
| s.d. of intercept ($S_a$) | 10.18 | 7.31 | 13.65 |
| s.d. of slope ($S_b$) | 0.25 | 0.24 | 0.45 |
| percentage relative standard deviation, % RSD | 1.40 | 0.50 | 0.73 |
| percentage relative error, % error | 0.57 | 0.20 | 0.30 |
| limit of detection, LOD (µg ml$^{-1}$) | 0.88 | 0.29 | 0.20 |
| limit of quantitation, LOQ (µg ml$^{-1}$) | 2.66 | 0.89 | 0.60 |

coefficients (*r*) were found to be 0.999 for the three drugs. Regression equations of the data were calculated and given as

$$y = -75.0650 + 38.35\,C \ (r = 0.9999) \text{ for FLU,}$$
$$y = 135.96 + 82.27\,C \ (r = 0.9999) \text{ for ITR,}$$
$$y = 62.5 + 229.14\,C \ (r = 0.9999) \text{ for TRH.}$$

**Table 5.** Assay results for the determination of the studied drugs in pure form by the proposed and comparison methods. (Values between parentheses are the tabulated $t$ and $F$ values, respectively, at $p = 0.05$ [42].)

| compound | proposed method | | | comparison methods [6,15,23] |
| | amount taken (µg ml$^{-1}$) | amount found (µg ml$^{-1}$) | % found | % found |
| --- | --- | --- | --- | --- |
| FLU | 5.00 | 5.09 | 101.72 | 100.76 |
| | 20.00 | 19.61 | 98.05 | 101.30 |
| | 25.00 | 24.90 | 99.61 | 98.83 |
| | 35.00 | 35.59 | 101.69 | 99.25 |
| | 40.00 | 39.89 | 99.74 | |
| | 80.00 | 79.92 | 99.90 | |
| mean | 100.12 | | | 100.04 |
| ± s.d. | 1.40 | | | 1.18 |
| $t$-test | 0.19 (2.37) | | | |
| $F$-test | 1.89 (5.79) | | | |
| ITR | 5.00 | 4.97 | 99.43 | 100.23 |
| | 10.00 | 9.97 | 99.67 | 100.65 |
| | 20.00 | 20.16 | 100.82 | 99.47 |
| | 30.00 | 29.95 | 99.82 | 97.87 |
| | 40.00 | 39.88 | 99.69 | am |
| | 50.00 | 50.07 | 100.15 | |
| mean | 99.93 | | | 99.56 |
| ± s.d. | 0.50 | | | 1.23 |
| $t$-test | 0. 3 (2.37) | | | |
| $F$-test | 2.97 (5.79) | | | |
| TRH | 1.00 | 0.99 | 98.84 | 98.52 |
| | 5.00 | 5.05 | 101.03 | 100.72 |
| | 20.00 | 20.04 | 100.19 | 100.00 |
| | 30.00 | 29.88 | 99.61 | 99.31 |
| | 40.00 | 39.95 | 99.87 | |
| | 50.00 | 50.09 | 100.19 | |
| mean | 99.96 | | | 99.64 |
| ± s.d. | 0.73 | | | 0.94 |
| $t$-test | 0.04 (2.37) | | | |
| $F$-test | 2.05 (5.79) | | | |

The limit of detection (LOD) and limit of quantitation (LOQ) were calculated according to the following equations [41], and the resulted data are shown in table 4:

$$LOQ = \frac{10S_a}{b},$$

$$LOD = \frac{3.3S_a}{b}.$$

By applying statistical analysis [42], no significant difference was found after comparing the results obtained from the proposed method with those from the comparison methods [6,9,10], as represented in table 5. The comparison methods were spectrophotometric methods that depend on measuring the

**Table 6.** Precision data for the determination of the studied drugs by the proposed HPLC method. (Note: each result is the average of three separate determinations.)

| drug | conc. (μg ml$^{-1}$) | intra-day | | | inter-day | | |
|---|---|---|---|---|---|---|---|
| | | mean ± s.d. | %RSD | % error | mean ± s.d. | %RSD | % error |
| FLU | 10.00 | 99.10 ± 0.50 | 0.41 | 0.24 | 99.77 ± 0.77 | 0.62 | 0.36 |
| | 20.00 | 99.60 ± 1.60 | 1.60 | 0.93 | 99.80 ± 0.20 | 1.12 | 0.65 |
| | 30.00 | 100.00 ± 1.70 | 1.89 | 1.09 | 99.50 ± 0.80 | 0.77 | 0.44 |
| ITR | 10.00 | 100.20 ± 1.00 | 1.25 | 0.72 | 100.10 ± 0.70 | 0.72 | 0.41 |
| | 20.00 | 99.70 ± 0.70 | 0.76 | 0.44 | 100.80 ± 0.80 | 0.82 | 0.48 |
| | 30.00 | 100.10 ± 0.10 | 0.12 | 0.07 | 100.00 ± 0.40 | 0.41 | 0.24 |
| TRH | 10.00 | 100.30 ± 1.30 | 1.36 | 0.78 | 99.40 ± 0.80 | 0.83 | 0.44 |
| | 20.00 | 100.40 ± 0.90 | 0.92 | 0.53 | 99.80 ± 0.50 | 0.51 | 0.29 |
| | 30.00 | 99.60 ± 0.60 | 0.74 | 0.43 | 100.25 ± 0.35 | 0.36 | 0.21 |

**Table 7.** Application of the proposed method for the determination of FLU, ITR and TRH in their pharmaceuticals.

| compound | proposed method | | | comparison method | | |
|---|---|---|---|---|---|---|
| | amount taken (μg ml$^{-1}$) | amount found (μg ml$^{-1}$) | % found | amount taken (μg ml$^{-1}$) | amount found (μg ml$^{-1}$) | % found |
| Flucoral® capsule | 10.00 | 9.92 | 99.20 | 150.00 | 150.84 | 100.56 |
| (150 mg/ | 20.00 | 20.15 | 100.80 | 200.00 | 198.34 | 99.17 |
| capsule) | 30.00 | 29.92 | 99.73 | 250.00 | 250.84 | 100.34 |
| mean | | | 99.91 | | | 100.02 |
| ± s.d. | | | 0.82 | | | 0.75 |
| t (2.78)* | | | 0.16 | | | |
| F (19)* | | | 1.20 | | | |
| Itrapex® capsule | 10.00 | 9.93 | 99.40 | 20.00 | 20.06 | 100.32 |
| (100 mg/ | 20.00 | 20.12 | 100.60 | 30.00 | 29.87 | 99.57 |
| capsule) | 30.00 | 23.93 | 99.80 | 40.00 | 40.06 | 100.16 |
| mean | | | 99.93 | | | 100.02 |
| ± s.d. | | | 0.61 | | | 0.40 |
| t | | | 0.18 | | | |
| F | | | 2.33 | | | |
| Lamisil® tablet | 10.00 | 10.09 | 100.98 | 10.00 | 10.05 | 100.6 |
| (250 mg/tablet) | 20.00 | 19.80 | 99.02 | 15.00 | 14.88 | 99.2 |
| | 30.00 | 30.09 | 100.33 | 20.00 | 20.05 | 100.3 |
| mean | | | 100.11 | | | 100.03 |
| ± s.d. | | | 1.00 | | | 0.74 |
| t | | | 0.12 | | | |
| F | | | 1.83 | | | |

absorbance of the three drugs in their solutions at 261.6 nm, 255 nm and 283 nm for FLU, ITR and TRH, respectively.

In order to determine intra-day and inter-day precision, three different concentrations of each drug were evaluated on three consecutive occasions between 1 day and three successive days. The analytical results are summarized in table 6.

**Table 8.** Application of the proposed HPLC method to the determination of the studied drugs in their synthetic mixtures.

| synthetic mixture | amt. taken ($\mu$g ml$^{-1}$) | | | % found | | |
|---|---|---|---|---|---|---|
| | FLU | ITR | TRH | FLU | ITR | TRH |
| | 10.00 | 20.00 | 50.00 | 98.06 | 98.74 | 98.14 |
| | 20.00 | 10.00 | 30.00 | 101.29 | 100.95 | 101.24 |
| | 50.00 | 40.00 | 10.00 | 99.87 | 99.84 | 99.63 |
| mean % | | | | 99.74 | 99.84 | 99.67 |
| $\pm$ s.d. | | | | 1.62 | 1.11 | 1.55 |

**Table 9.** Assay results for the determination of the studied drugs in spiked human plasma samples using the proposed method.

| | amount found ($\mu$g ml$^{-1}$) | | amount taken ($\mu$g ml$^{-1}$) | | % found | |
|---|---|---|---|---|---|---|
| | ITR | TRH | ITR | TRH | ITR | TRH |
| spiked | 5.00 | 0.50 | 5.27 | 0.52 | 105.5 | 104.6 |
| human | 10.00 | 1.00 | 9.54 | 1.01 | 95.41 | 101.0 |
| plasma | 17.00 | 1.30 | 17.19 | 1.23 | 101.15 | 95.08 |
| | 20.00 | 2.00 | 20.01 | 2.03 | 100.09 | |
| mean | | | | | 100.54 | 100.56 |
| $\pm$ s.d. | | | | | 4.14 | 3.98 |
| %RSD | | | | | 4.12 | 3.96 |
| % error | | | | | 2.06 | 1.98 |

The robustness of the proposed HPLC methods has been proved by making a slight variation in the chromatographic conditions such as pH ($7 \pm 0.2$), methanol ($95 \pm 2\%$) and flow rate ($0.7 \pm 0.1$ min ml$^{-1}$). Such minor changes were found to have minimal effect on drug resolution, indicating good robustness of the method.

## 3.3. Applications

### 3.3.1. Application in tablets dosage forms and synthetic mixtures

The results obtained from the suggested method for evaluating FLU, ITR and TRH in commercial tablets and synthetic mixtures were compared with those obtained using previous methods [6,9,10]. By applying statistical analysis using Student's $t$-test and variance ratio $F$-test [42], no important difference regarding the accuracy and precision was found, as shown in tables 7 and 8.

### 3.3.2. Application in biological fluid

The suggested approach applied to determine ITR and TRH in spiked human plasma. The results obtained from spiked plasma are presented in table 9. Using the optimized experimental conditions, a linear relationship was constructed by plotting the peak area against the drug concentration.

Therapeutic concentration of ITR is greater than 0.25 mg l$^{-1}$ and peak plasma levels of approximately 1 mg l$^{-1}$ TRH occur 2 h after a single oral dose of 250 mg [43] and other reports indicate higher plasma concentration of ITR greater than 1.0 mg ml$^{-1}$ in resistant oropharyngeal candidiasis [44].

Linear regression analysis of the data gave the following equation:

$$PA = 562.14 + 61.1 \, C \; (r = 0.9932) \text{ for ITR,}$$
$$PA = 271.4 + 513.4 \, C \; (r = 0.9976) \text{ for TRH.}$$

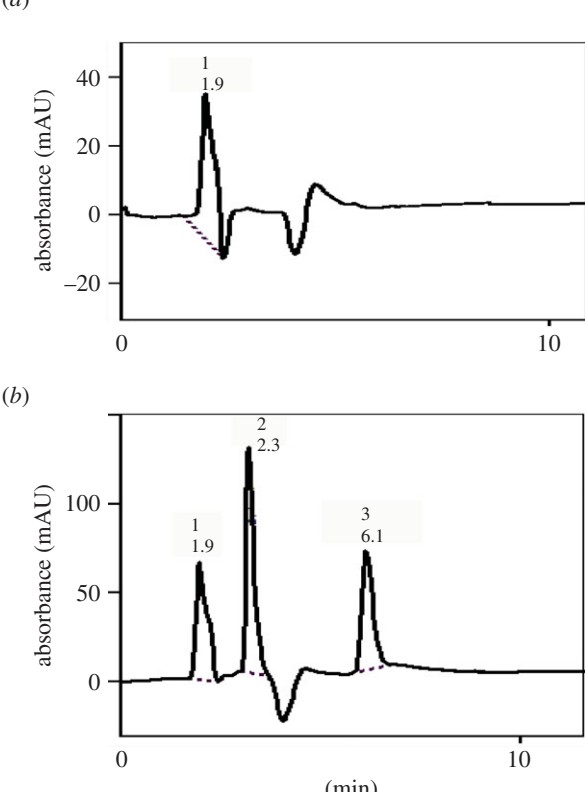

**Figure 8.** Typical chromatogram of the studied drugs in spiked human plasma under the described chromatographic conditions. (*a*) Plasma blank and (*b*) 1, solvent front; 2, ITR (17 µg ml$^{-1}$); and 3, TRH (1.7 µg ml$^{-1}$).

Fluconazole could not be determined in spiked human plasma by this method because of overlapping of its peak with the plasma peak, as shown in figure 8.

# 4. Conclusion

A factorial design-assisted HPLC method has been developed to separate three antifungal drugs, namely FLU, ITR and TRH. The proposed method is a straightforward one that saves time and money for reaching the optimum chromatographic conditions. The developed methodology permitted the whole separation to proceed in less than 8 min. Moreover, both ITR and TRH can be determined in spiked human plasma with satisfactory results. The proposed methodology is fast, simple and reproducible with a wide linear range in comparison with other previous reports for the analysis of the three drugs. It could be used for routine analysis of these antifungal drugs.

Ethics. Collection of biological samples has been reviewed and reapproved by the ethical committee of Faculty of Pharmacy, Mansoura University (code no. 2020-99).
Data accessibility. Data are available from the Dryad Digital Repository: https://doi.org/10.5061/dryad.sqv9s4n2t [45].
Authors' contributions. Laboratory work was carried out by A.R. in addition to data analysis and drafting the manuscript. H.E. and S.S. contributed to the study design and the statistics. A.E.-B. organized the overall study. All authors permitted the publication of the manuscript.
Competing interests. We declare we have no competing interests.
Funding. We received no funding for this study.

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
