## [Peer Review File · Royal Society Open Science]

Review History

RSOS-202130.R0 (Original submission)

Review form: Reviewer 1

Is the manuscript scientifically sound in its present form?

Yes

Are the interpretations and conclusions justified by the results?

Yes

Is the language acceptable?

No

Do you have any ethical concerns with this paper?

No

Have you any concerns about statistical analyses in this paper?

No

Recommendation?

Accept with minor revision (please list in comments)

Comments to the Author(s)

The manuscript is fine and can be accepted as is, however I advise the authors to check and revise the language for any mistakes.

Review form: Reviewer 2

Is the manuscript scientifically sound in its present form?

Yes

Are the interpretations and conclusions justified by the results?

Yes

Is the language acceptable?

Yes

Do you have any ethical concerns with this paper?

No

Have you any concerns about statistical analyses in this paper?

No

Recommendation?

Accept with minor revision (please list in comments)

Comments to the Author(s)

The manuscript could be accepted after minor revision:

1. Advantage of the proposed method over the published ones should be introduced in the introduction
2. Significance for determination of the three drugs simultaneously should be presented
3. In the selection of wavelength: what is the Lambda max. for each of the three drugs that you based your selection on. A figure of the absorption spectra should be added. Does detection at 210 nm cause any interference in measurements?
4. Please revise the grammar and spelling mistakes in the abstract
5. In results and discussion, please add references to the Fisher Statistical Test for Variance Analysis (ANOVA) model and The polynomial equation
6. Please revise the adjustment of the text within the manuscript to be uniform
7. Reduce the theoretical part about DOE, it is well established
8. Figures from 3 to 7 you can combine each set of figures as inset, for example figure 3 contain 4 subfigures combine them
9. Please revise the significant figures all over the manuscript

Review form: Reviewer 3

Is the manuscript scientifically sound in its present form?

Yes

Are the interpretations and conclusions justified by the results?

Yes

Is the language acceptable?

Yes

Do you have any ethical concerns with this paper?

No

Have you any concerns about statistical analyses in this paper?

No

Recommendation?

Accept with minor revision (please list in comments)

Comments to the Author(s)

No. of keywords is large

In the abstract; the no. of significant figures for QL and DL is not consistent.

The solutions were then filtered to get clear. In section 2.3.1. complete the sentence:

choice of column: clarify the results of the separation for the other two columns.

biological fluids; pharmaceutical dosage forms should be removed from the key words.

The detection limit (DOL) and The quantitation limit (QOL): DL and QL or limit of detection and limit of quantitation

Check the references for extra commas, question marks and style of the journal.

In table 3; mention the pharmacopeial values for the measured parameters as a footnote

In tables (5-8) add at least one significant figure to all mentioned concentrations in the amount taken column

All significant figures should be unified in all tables

Decision letter (RSOS-202130.R0)

This year has been very difficult for everyone, and we want to take the opportunity to thank you for your continued support in 2020.

The Royal Society Open Science editorial office will be closed from the evening of Friday 18 December 2020 until Monday 4 January 2021. We will not be responding during this time. If you have received a deadline within this time period, please contact us as soon as possible to allow us to extend the deadline. If you receive any automated messages during this time asking you to meet a deadline, we offer apologies and invite you to respond after the festive period or during normal working hours.

With our best for a peaceful festive period and New Year, and we look forward to working with you in 2021.

Dear Dr Fouda:

Title: Factorial design assisted RP-HPLC method for simultaneous determination of fluconazole, itraconazole and terbinafine.

Manuscript ID: RSOS-202130

Thank you for submitting the above manuscript to Royal Society Open Science. On behalf of the Editors and the Royal Society of Chemistry, I am pleased to inform you that your manuscript will be accepted for publication in Royal Society Open Science subject to minor revision in accordance with the referee suggestions. Please find the reviewers' comments at the end of this email.

The reviewers and handling editors have recommended publication, but also suggest some minor revisions to your manuscript. Therefore, I invite you to respond to the comments and revise your manuscript.

Because the schedule for publication is very tight, it is a condition of publication that you submit the revised version of your manuscript before 01-Jan-2021. Please note that the revision deadline will expire at 00.00am on this date. If you do not think you will be able to meet this date please let me know immediately.

Supplementary files will be published alongside the paper on the journal website and posted on the online figshare repository (<https://figshare.com>). The heading and legend provided for each supplementary file during the submission process will be used to create the figshare page, so please ensure these are accurate and informative so that your files can be found in searches. Files

on figshare will be made available approximately one week before the accompanying article so that the supplementary material can be attributed a unique DOI.

Kind regards,
Dr Laura Smith
Publishing Editor, Journals

RSC Associate Editor:
Comments to the Author:
(There are no comments.)

RSC Subject Editor:
Comments to the Author:
(There are no comments.)

Reviewer comments to Author:
Reviewer: 1

Comments to the Author(s)
The manuscript is fine and can be accepted as is, however I advise the authors to check and revise the language for any mistakes.

Reviewer: 2

Comments to the Author(s)
The manuscript could be accepted after minor revision:

1. Advantage of the proposed method over the published ones should be introduced in the introduction
2. Significance for determination of the three drugs simultaneously should be presented
3. In the selection of wavelength: what is the Lambda max. for each of the three drugs that you based your selection on. A figure of the absorption spectra should be added. Does detection at 210 nm cause any interference in measurements?
4. Please revise the grammar and spelling mistakes in the abstract
5. In results and discussion, please add references to the Fisher Statistical Test for Variance Analysis (ANOVA) model and The polynomial equation

6. Please revise the adjustment of the text within the manuscript to be uniform
7. Reduce the theoretical part about DOE, it is well established
8. Figures from 3 to 7 you can combine each set of figures as inset, for example figure 3 contain 4 subfigures combine them
9. Please revise the significant figures all over the manuscript

Reviewer: 3

Comments to the Author(s)

No. of keywords is large

In the abstract; the no. of significant figures for QL and DL is not consistent.

The solutions were then filtered to get clear. In section 2.3.1. complete the sentence:

choice of column: clarify the results of the separation for the other two columns.

biological fluids; pharmaceutical dosage forms should be removed from the key words.

The detection limit (DOL) and The quantitation limit (QOL): DL and QL or limit of detection and limit of quantitation

Check the references for extra commas, question marks and style of the journal.

In table 3; mention the pharmacopeial values for the measured parameters as a footnote

In tables (5-8) add at least one significant figure to all mentioned concentrations in the amount taken column

All significant figures should be unified in all tables

Author's Response to Decision Letter for (RSOS-202130.R0)

See Appendix A.

Decision letter (RSOS-202130.R1)

Dear Dr Fouda:

Title: Factorial design assisted RP-HPLC method for simultaneous determination of fluconazole, itraconazole and terbinafine.

Manuscript ID: RSOS-202130.R1

It is a pleasure to accept your manuscript in its current form for publication in Royal Society Open Science. The chemistry content of Royal Society Open Science is published in collaboration with the Royal Society of Chemistry.

RSC Associate Editor
Comments to the Author:
(There are no comments.)

Reviewer(s)' Comments to Author:

Appendix A

On behalf of my co-authors, I would like to thank the editorial team for giving us further chance to revise our manuscript. All the reviewer comments were taken into consideration.

Reviewer 1:

- The manuscript is fine and can be accepted as is, however I advise the authors to check and revise the language for any mistakes.

Reply: The language was further revised and corrected.

Reviewer 2:

1. Advantage of the proposed method over the published ones should be introduced in the introduction

Reply: The advantages of the proposed method was abridged.

2. Significance for determination of the three drugs simultaneously should be presented

Reply: The significance is related to the presence of evidence that the combination of these three antifungal drugs is recommended for treatment of *Aspergillus*, *Candida*, *Mucorales* species and against fluconazole-resistant *Candida* isolates and itraconazole-resistant *Aspergillus* strains with minimal side effects and high efficacy (36).

3. In the selection of wavelength: what is the Lambda max. for each of the three drugs that you based your selection on. A figure of the absorption spectra should be added. Does detection at 210 nm cause any interference in measurements?

Reply: No interference was observed as indicated from the analysis of different pharmaceutical preparations and biological fluids.

The figure was inserted as Fig.2.

4. Please revise the grammar and spelling mistakes in the abstract

Reply: The language was checked and corrected.

5. In results and discussion, please add references to the Fisher Statistical Test for Variance Analysis (ANOVA) model and The polynomial equation

Reply: Reference was added as recommended by the reviewer.

6. Please revise the adjustment of the text within the manuscript to be uniform

Reply: The text was adjusted as required

7. Reduce the theoretical part about DOE, it is well established

Reply: The required modification was carried out

8. Figures from 3 to 7 you can combine each set of figures as inset, for example figure 3 contain 4 subfigures combine them

Reply: The required changes were carried out

9. Please revise the significant figures all over the manuscript

Reply: Significant figures were revised and checked all over the manuscript

Reviewer 3:

Comments to the Author(s)

- No. of keywords is large

Reply: The keywords No. was reduced

- In the abstract, the no. of significant figures for QL and DL is not consistent

Reply: The No. was unified

- The solutions were then filtered to get clear. In section 2.3.1. complete the sentence:

Reply: The sentence has been completed.

- choice of column: clarify the results of the separation for the other two columns.

Reply: The choice of column has been discussed as instructed

- biological fluids; pharmaceutical dosage forms should be removed from the key words.
- The detection limit (DOL) and The quantitation limit (QOL): DL and QL or limit of detection and limit of quantitation

Reply: the required changes have been carried out

- Check the references for extra commas, question marks and style of the journal.

Reply: The references have been revised carefully.

- In table 3; mention the pharmacopeial values for the measured parameters as a footnote

Reply: the required changes have been carried out

- In tables (5-8) add at least one significant figure to all mentioned concentrations in the amount taken column

Reply: the concentrations have been corrected.

- All significant figures should be unified in all tables

Reply: As recommended, all significant figures were checked.